# Seeing Beyond Words: Multimodal Aspect-Level Complaint Detection in Ecommerce Videos

## ABSTRACT

Complaints are pivotal expressions within e-commerce communication, yet the intricate nuances of human interaction present formidable challenges for AI agents to grasp comprehensively. While recent attention has been drawn to analyzing complaints within a multimodal context, relying solely on text and images is insufficient for organizations. The true value lies in the ability to pinpoint complaints within the intricate structures of discourse, scrutinizing them at a granular aspect level. Our research delves into the discourse structure of e-commerce video-based product reviews, pioneering a novel task we term Aspect-Level Complaint Detection from Discourse (ACDD). Embedded in a multimodal framework, this task entails identifying aspect categories and assigning complaint/non-complaint labels at a nuanced aspect level. To facilitate this endeavour, we have curated a unique multimodal product review dataset, meticulously annotated at the utterance level with aspect categories and associated complaint labels. To support this undertaking, we introduce a Multimodal Aspect-Aware Complaint Analysis (MAACA) model that incorporates a novel pre-training strategy and a global feature fusion technique across the three modalities. Additionally, the proposed framework leverages a moment retrieval step to identify the relevant portion of the clip, crucial for accurately detecting the fine-grained aspect categories and conducting aspect-level complaint detection. Extensive experiments conducted on the proposed dataset showcase that our framework outperforms unimodal and bimodal baselines, offering valuable insights into the application of video-audio-text representation learning frameworks for downstream tasks. *The code and the sample dataset are shared as Supplementary Material.*

## KEYWORDS

Aspect-based Multimodal Complaint Detection, Multimodal Fusion, Video-Audio-Text Alignment, Multi-task Learning, Social Media Mining, Multimedia Applications

## 1 INTRODUCTION

In today's landscape, the surge in product review videos across platforms like YouTube[1], seamlessly integrated into major e-commerce hubs such as Amazon[2], highlights the escalating impact of visual

---

[1]https://www.youtube.in
[2]https://www.amazon.in

content on consumer choices. As these multimedia reviews increasingly shape purchasing decisions, understanding user sentiments and pinpointing specific complaint aspects within these videos has become a crucial focus [15, 28, 32]. However, current research largely overlooks the immense potential lying at the intersection of visual, audio, and textual data within these video reviews.

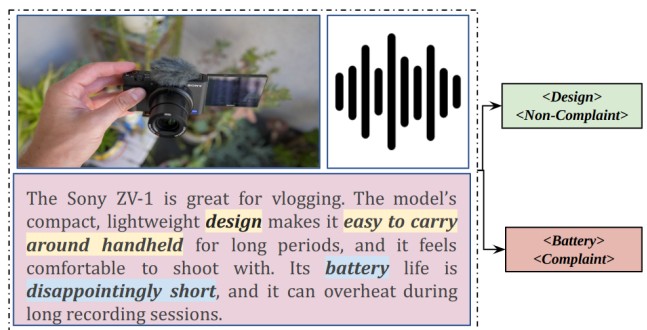

The Sony ZV-1 is great for vlogging. The model's compact, lightweight **design** makes it **easy to carry around handheld** for long periods, and it feels comfortable to shoot with. Its **battery** life is **disappointingly short**, and it can overheat during long recording sessions.

**Figure 1: An example of aspect-based complaint detection using multimodal cues.**

**Motivation:** The significance of incorporating visual elements in complaint detection becomes apparent due to several reasons. Individuals expressing complaints often share information using visual and audio formats alongside text, which is crucial for providing accurate details about various aspects that trigger complaints. Moreover, aspects like reviewer's tone or electronic design often necessitate audio and visual cues for precise identification, surpassing what text alone can convey. Merging text, visuals, and audio in fine-grained complaint detection has the potential to enhance accuracy and efficiency, offering a more comprehensive understanding that solely textual analysis may overlook. This strategy acknowledges the intricate ways in which humans express themselves, leveraging multimodal information in their communication. By delving into the integration of text, images, and audio, computational linguistics researchers can better tackle the evolving complexities of modern communication.

This interdisciplinary approach to complaint detection holds numerous advantages for both companies and consumers. For companies, it offers deeper consumer insights, nuanced aspect identification, improved user experience design, precise marketing strategies, and fosters trust-building communication. On the consumer side, it empowers them with a clearer voice, ensures a more accurate representation of their concerns, and helps in forming realistic expectations regarding products and services. Ultimately, this convergence of visual, audio, and textual data not only enriches the understanding of consumer feedback but also facilitates more informed decisions, leading to improved products and enhanced user experiences.

However, the lack of effective, automated methodologies for systematically analyzing the diverse range of video content and identifying complaints at the aspect level poses a substantial challenge. Moreover, the scarcity of datasets for analyzing complaint entities in video reviews emphasizes the need to develop comprehensive resources in this domain. This motivates our research to curate a dataset of video reviews and study various approaches involving multiple modalities for fine-grained complaint recognition.

In response, the Aspect-Level Complaint Detection from Discourse framework includes two sub-tasks: aspect category detection (ACD), and aspect-level complaint classification (ACC) using the textual, visual, and audio data available. The initial focus is on pinpointing aspects within instances and categorizing them into specific aspect categories. Subsequently, the task involves whether instances at the aspect level constitute complaints or non-complaints. As shown in Figure 1, for a given multimodal review consisting of textual, visual and audio data, the two identified duplets (aspect category-complaint label) are shown on the right side.

**Contributions:** The major contributions of this work include:

- We introduce the Video Complaint Dataset (VCD), a novel resource aimed at advancing research in aspect-level complaint detection.
- We propose a Multimodal Aspect-Aware Complaint Analysis (MAACA) framework for aspect-level complaint detection from discourse (ACDD). MAACA extends the ALPRO pre-training strategy [18], to incorporate the audio modality into its architecture as well as in its pre-training strategy. Furthermore, MAACA incorporates a moment retrieval step, augmenting the identification of pertinent segments within the video clip crucial for the accurate detection of fine-grained aspect categories and aspect-level complaints.
- We propose a gated-fusion mechanism to efficiently integrate multimodal representations while considering the varying importance of each feature through a gating mechanism.
- Extensive experiments conducted on the VCD dataset demonstrate the significant superiority of our framework over existing multimodal baselines, providing valuable insights into the application of multimodal representation learning frameworks for downstream tasks.

## 2 RELATED WORK

**Complaint and Text:** In the realm of linguistics and psychology, it's consistently observed that people adapt their complaints to various degrees [12, 22, 37]. Complaints can either be implicit, without assigning blame, or explicit, directly accusing someone of wrongdoing [38]. These complaints are further categorized by emotional intensity into four levels: no particular blame, disapproval, accusation, and blame [37]. Minor complaints can serve as emotional outlets and enhance mental well-being, whereas severe complaints can lead to hostility and aggression [11].

In computational linguistics, previous research has predominantly focused on automating the identification of text-based review-level complaints [5, 13, 28]. Jin and Aletras [13] employed transformer-based models and linguistic data to assess the seriousness of complaints and predict their severity. Furthermore, multitask complaint

analysis models that incorporate sentiment and emotional information have been developed specifically for text-based content [35, 36]. Recently, two new tasks, complaint cause detection and extraction, were introduced in [34], aiming to detect and extract the reasons behind Twitter complaints, introducing an interpretability dimension in complaint detection. Additionally, prior research has categorized complaints based on factors such as the responsible department, urgency, product hazards, and risks [3, 16, 39].

**Complaint and Multimodality:** The study discussed in [27, 31] has contributed to linking vision and language in the related area of polarity and emotion recognition. The study in [32] proposed a binary complaint classifier based on text and image information without considering the particular features or aspects about which the user is complaining. The work also publicly released a multimodal complaint dataset (CESAMARD) [32], a collection of consumer feedback or reviews and images of products purchased from the e-commerce website Amazon, which has aided additional investigations into complaint detection in multimodal setup. This dataset was further used for developing a fine-grained aspect-based complaint detection model in the work [33]. They proposed a multimodal-bitransformer-based architecture where in the first phase the aspect category is identified and then in the next phase the aspect-category complaint detection is performed using self-attention and BiGRU layers.

**Vision-and-Language Models:** In recent years, significant advancements have been made in training models to comprehend both vision and language simultaneously, leveraging vast multimodal datasets [4, 19, 20]. These models integrate both modalities into a unified input and are trained using objectives akin to masked language modelling. However, most existing methods for text-video retrieval [8, 23] rely on pre-trained visual backbones, which densely extract video features for each frame offline. Nevertheless, since these visual backbones are typically pre-trained on image and/or video datasets without textual information, their features are less effective for tasks involving both video and language.

Recent approaches such as ClipBERT [17] have shown promising results by fine-tuning the visual backbone end-to-end using only a small number of sparsely sampled frames. However, since ClipBERT is pre-trained on image-text data, it struggles to effectively aggregate information across frames. In contrast, ALPRO [18], a video-language pre-training method, learns robust cross-modal representations from sparse video frames and accompanying texts. Additionally, by leveraging instance-level video-text alignment, ALPRO can generate accurate entity pseudo-labels with a wide vocabulary, thereby enhancing the efficiency and effectiveness of region-entity alignment learning. This motivates us to take advantage of the ALPRO pre-training framework incorporated with audio cues specifically for the ACDD task, improving the performance specifically.

In recent years, while progress has been made in detecting text and image-based complaints, the exploration of video-based complaint detection remains largely uncharted. Videos offer a wealth of information, incorporating visual and auditory cues that enable a more comprehensive expression of concerns and grievances. Unlike text and images, videos capture not only the content but also the manner of delivery, providing nuanced details. This paper addresses

**Table 1: Annotation guidelines for *VCD* dataset.**

| S.No. | Annotation Guidelines |
|-------|----------------------|
| 1 | The aspect category should consider the complainant's perspective. |
| 2 | Each selected aspect category should refer to either a complaint or non-complaint. |
| 3 | Each aspect category should be strictly marked with the start and end time stamps. |
| 4 | In case of confusion regarding annotation it should be reported and rectified. |

the need to fill this research gap with video-based complaint detection at the fine-grained aspect level by introducing a novel problem statement and dataset. It aims to foster the development of effective models and techniques for identifying, categorizing, and analyzing complaints within video content.

## 3 VIDEO COMPLAINT DATASET (VCD)

In this segment, we delve into the gathering of data and annotations concerning different attributes at the level of individual utterances. These attributes encompass aspect categories, complaint and non-complaint labels at the aspect level, and commencement and cessation timestamps.

### 3.1 Data Collection and Processing

We gathered a total of 130 review videos from YouTube, with a specific focus on electronic products, including phones, laptops, and cameras, with 450 total annotated utterances. Each utterance within a discourse-oriented video is identified by its start and end timestamps, defined as a segment of speech demarcated by breaths or pauses [9]. Breaking down the data further, 95 reviews pertain to the phone domain, 22 to the laptop domain, and 13 to the camera domain. This distribution allows for a comprehensive exploration of customer perspectives across different electronic product categories. Transcripts are created manually for every video in cases where they are not provided on YouTube.

The strategic focus on the domains of laptops, phones, and cameras was driven by several compelling reasons. Firstly, these three domains represent crucial segments within the broader electronics industry, which is currently experiencing rapid growth and technological advancements. By centring our collection efforts on these three domains, we aimed to capture and analyze the evolving consumer sentiments and preferences in these pivotal areas. Moreover, the decision to concentrate on these specific domains was influenced by the widespread availability and popularity of gadget reviews on YouTube, with electronic products garnering significant attention from consumers seeking informed purchasing decisions. Gadget reviews, particularly those related to phones, laptops, and cameras, are highly sought after due to the constant innovation and updates in these technologies. The extensive user-generated content and discussions around these products on YouTube make it an ideal platform for extracting diverse and valuable insights.

### 3.2 Annotator Details

In this study, we enlisted three annotators to evaluate aspect-level complaints in our dataset. The selection process for these annotators involved a competitive screening, open exclusively to students from the computer science department with expertise in natural language processing. After providing them with the annotation guidelines

(Table 1), they annotated a set of 20 video samples for aspect classes and complaint/non-complaint labels. Based on the quality, accuracy, and semantic coherence of their annotations, we selected three students to annotate the video complaint dataset. Among them, two hold Ph.D. qualifications, while the third is a post-graduate student. All three are skilled in labelling tasks and possess a deep understanding of the subject area, along with significant experience in constructing supervised datasets. Notably, they are proficient in English, having pursued their education in an English-based academic environment.

### 3.3 Annotation Phase & Dataset Analysis

The annotators were provided with the annotation guidelines outlined in Table 1 and a set of 20 annotated video samples for reference. This approach aimed to support the annotators during annotation and to help resolve any uncertainties that might arise. Our annotation methodology draws inspiration from previous studies in aspect-based sentiment analysis, particularly SemEval shared challenges [24–26]. Within the electronics domain, various relevant aspect categories were identified, including camera, OS, design, battery, price, speaker, and storage. Annotators were tasked with identifying the appropriate aspect category that best correlated with the issue discussed in the review. This involved meticulous examination of text, video, and audio data to pinpoint the aspect category and its corresponding complaint/non-complaint label for each speaker utterance.

The final aspect classes and corresponding complaint/non-complaint labels were determined through majority voting. In cases where the annotations differed among the three annotators, the authors worked to resolve any ambiguity or uncertainty. We calculated the Fleiss-Kappa [7] agreement scores to assess the overall agreement among raters, a commonly used method for cases with more than two annotators. The aspect category detection (ACD) and aspect complaint classification (ACC) tasks yielded scores of 0.76 and 0.84, respectively, indicating significant agreement among annotators for both tasks [1]. Table 2 illustrates a few examples with the annotated aspect categories and the corresponding aspect-level complaint labels sourced from the *VCD* dataset. *Kindly refer to Supplementary Material for more details regarding the VCD dataset.*

## 4 METHODOLOGY

In this section, we present the technical aspects of the proposed framework. We begin by defining the problem, followed by an in-depth examination of the overall architecture, depicted in Figure2.

### 4.1 Problem definition

Each data point consists of a video review consisting of a transcript as textual data T, Video V as per the time stamp $t_{ij}$, and audio clip

**Table 2: Examples from the *VCD* dataset with ACD, ACC task annotation.**

| Review | Video | Aspect Categories | Label |
|---|---|---|---|
| I simply did not like the audio output; from the headphone jack. If you listen the sound output is very flat and sort of lifeless. | | Speaker | Complaint |
| The device has Gorilla Glass 3 protection, ensuring good screen quality. Battery life appears to be commendable lasting around 3 days. But the front-facing camera, rated at 12 megapixels, offers only average performance. | | Design

Battery
Camera | Non-Complaint

Non-Complaint
Complaint |

A, related to the electronic item being discussed in the video. The ultimate output is a pair that includes both the aspect category and the corresponding complaint or non-complaint label, identified through the analysis of the dataset's three modalities.

## 4.2 Moment-Retrieval of Relevant Frames

Accurate identification of relevant segments within review videos is crucial for the task of aspect-level complaint detection from discourse (ACDD), particularly since we are dealing with videos with inherent noise. Review videos often include extended introductions, outros, or cutaway shots of the reviewer's face (as shown in Table 2), which hold no value for classifying the complaint or identifying the aspect. Focusing on relevant segments allows for the extraction of more concise visual features for classification.

Thus, we propose moment retrieval as an important pre-processing step to enhance the model's performance. Moment retrieval is the task of localizing the set of most relevant moments in an untrimmed video according to the given natural language query. To identify the relevant parts of the clip, we make use of CG-DETR [21], which achieves state-of-the-art results in QVHighlights dataset [30] created for moment-retrieval. CG-DETR leverages the correlation between the video and query to predict the saliency score for each frame denoting the likelihood of relevancy.

Each clip is passed into the CG-DETR model with a set of simple prompts denoting all the product categories in the dataset such as "Phone", "Laptop", "Camera", etc. Then, the time frame corresponding to the highest saliency score across all the prompts is taken. Thus we retrieve the section of the clip where the product under review is shown, thereby offering more relevant signals for complaint and aspect classification. The comparison of using the moment-retrieved clip and the entire clip as the input to the proposed model is discussed in the ablation study, kindly refer to subsection 5.3.1.

## 4.3 Framework Components

As illustrated in Figure 2, the model consists of three uni-modal encoders to obtain the feature representation of each modality and two multi-modal encoders to capture the interaction between the modalities. Also, other components such as Information Compression and Gated Multimodal Fusion modules are present, the details of which will be outlined in the following sections.

(1) **Text encoder:** We use two instances of the 6-layer BERT-encoders $E_{tv}$ and $E_{ta}$[6] to represent the text embedding $Z_{tv}$ and $Z_{ta}$, for video feature alignment and audio feature alignment respectively. The transcript $X_t$ corresponding to the clip is tokenized and truncated/padded up to $N_t = 128$ tokens and passed as inputs to the BERT-encoder. The value for $N_t$ was chosen upon careful consideration of the trade-off between the time complexity of attention in the further steps and the length of the transcript. Both the encoders output an embedding sequence of $t = \{t_{CLS}, t_1, ...t_{N_t}\}$ with $t_i \in R^D$ where D is the text embedding dimension.

$$Z_{tv} = E_{tv}(X_t) \tag{1}$$

$$Z_{ta} = E_{ta}(X_t) \tag{2}$$

(2) **Video encoder:** We employ TimeSFormer [2] as the video modality encoder $E_v$. The moment-retrieved clip $X_v$ is further sampled uniformly into T=16 frames of size 224x224x3. Each frame is chunked into P patches, flattened, and mapped to an embedding $z_{tp} \in R^D$ by a linear transformation. Layers of multi-headed self-attention are applied along the temporal and spatial dimensions of the patch embedding independently and combined to form patch embedding for each frame $\alpha_{tp} \in R^D$. The frame-level features are pooled across the temporal dimension to obtain video features $Z'_v$ of dimension $v \in R^{N'_v x D_v}$. where $D_v$ is the video embedding dimension.

$$Z'_v = E_v(X_v) \tag{3}$$

(3) **Audio encoder:** We leverage whisper-small [29] encoder $E_a$ to extract the audio features $Z'_a$. Audio corresponding to the text transcript is sampled at the rate of 16000 Hz and passed to the Whisper encoder. Input audio is split into 30-second chunks, converted into a log-Mel spectrogram, and then passed into an encoder consisting of a series of self-attention blocks. The output of the audio encoder has a dimension of $a \in R^{N_a x D_a}$ representing the audio features $Z'_a$, where $D_a$ is the audio embedding dimension.

$$Z'_a = E_a(X_a) \tag{4}$$

(4) **Information compression Bi-LSTM module:** Two instances of this Bi-LSTM [10] module exist, one corresponding to the Video (v) and the other to the Audio (a) modalities. The primary objective of this module is to condense the representation of the input sequences $Z'_v$ and $Z'_a$, while retaining the essential information specific to each modality. For a given modality $m \in \{a, v\}$, the input sequence

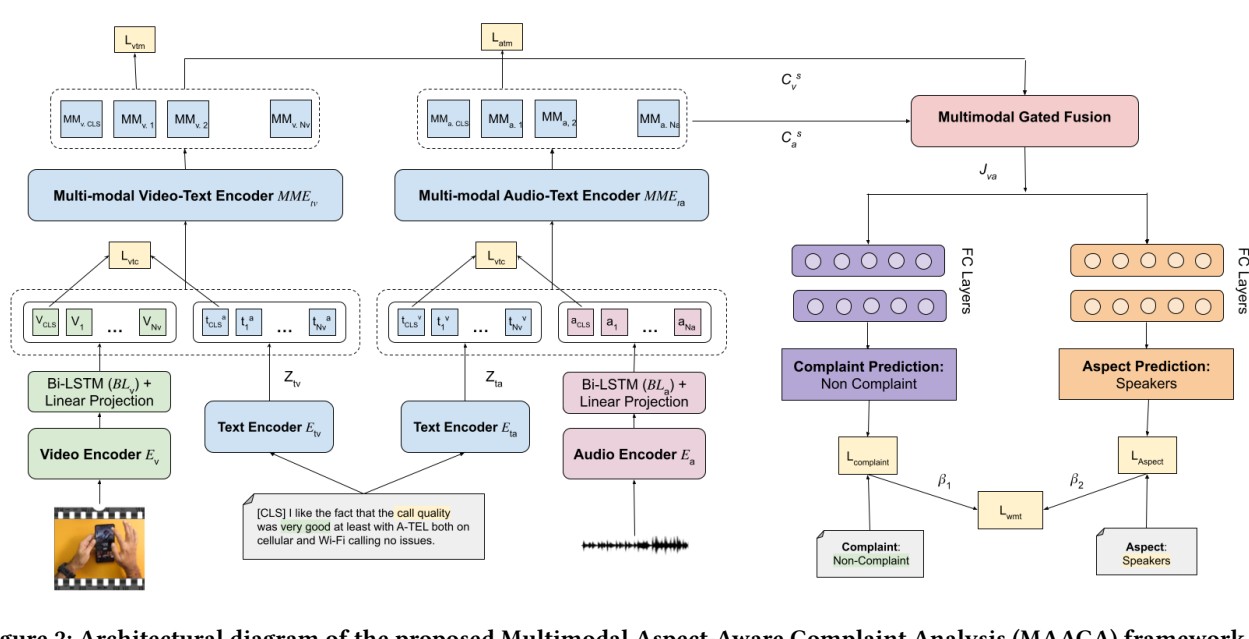

**Figure 2: Architectural diagram of the proposed Multimodal Aspect-Aware Complaint Analysis (MAACA) framework**

$Z'_m$ is fed into the respective Bidirectional Long Short-Term Memory (Bi-LSTM) module $BL_m$, resulting in an output sequence $Z_m$. This output sequence consists of the last $N_m$ timesteps of the modality-specific Bi-LSTM. The rationale behind this approach stems from the observation that the information encapsulated within an input sequence tends to be concentrated in the latter part of the sequence. These final $N_m$ timesteps are particularly significant as they are exposed to the cumulative information of the preceding timesteps, thereby inherently encoding the essence of the entire sequence up to that point. This approach effectively compresses the information while preserving the salient features of each modality-specific input sequence. The shape of the output sequence is $R^{N_m x D_m}$, where $N_m < N'_m$ and $m$ is the corresponding modality.

$$Z_m = BL_m(Z'_m), m \in \{a, v\} \quad (5)$$

.

(5) **Fully connected layer for embedding dimension alignment:** This linear projection layer is used to change the embedding dimension of $Z_m$ from $D_m$ to $D$. Where $m$ is the corresponding modality ($m \in \{a, v\}$), $D_m$ and $Z_m$ are corresponding modality-specific embedding dimensions and compressed feature representations, respectively. The dense layer for embedding dimension alignment operates by transforming the input feature representation $Z_m$ of modality $m$ to a new feature representation $Y_m$ using a linear projection. Thus producing the final audio and video features as $Y_m = \{m_{CLS}, m_1, ...m_{N_m}\}$ with $m_i \in R^D$, where $m \in \{a, v\}$.

(6) **Multimodal encoder module:** There are two instances of this module one each for video-text and audio-text pairs, denoted as $MME_{tv}$ and $MME_{ta}$ respectively (denoting multimodal encoders for both pairs). This encoder module comprises a 6-layer BERT encoder that employs self-attention

to encode the two-modality combinations. We concatenate the text representation $Z_{ta}$ and $Z_{ta}$ with the compressed, dimension-aligned video $Y_v$ and audio $Y_a$ representations to produce $Y_{tv}$ and $Y_{ta}$. This concatenated representation is passed through the 6-layer BERT encoder, to generate a unified multimodal representation through repeated self-attention modules for each pair given by $C^s_m$ where $m \in \{a, v\}$.

$$C^s_a = MME_{ta}(Y_{ta}) \quad (6)$$

$$C^s_v = MME_{tv}(Y_{tv}) \quad (7)$$

Here $Y_m$ and $C^s_m$ are given as a sequence of multimodal (MM) tokens $MM_{m,CLS}, MM_{m,1}, ...MM_{m,N_m+N_t}$ with each token $MM_i \in R^D$, where $m \in \{a, v\}$.

(7) **Gated multimodal fusion:** To combine text-aligned video and text-aligned audio tokens, we employ a gated attention strategy. Unlike directly assigning weights to each vector, the gate fusion mechanism enables varying contributions to the prediction from different positions of vectors. The joint representation resulting from the gate fusion is computed as follows:

$$\alpha = \sigma(\mathbb{P}_v C^s_v + \mathbb{P}_a C^s_a + b_g),$$
$$J_{va} = \alpha C^s_a + (1 - \alpha)C^s_v \quad (8)$$

Here, $\mathbb{P}_v$ and $\mathbb{P}_a$ represent weight matrices for the visual and acoustic modalities, while $b_g$ denotes scalar bias and $\sigma$ is the sigmoid activation function.

## 4.4 Pretraining:

In this section, we outline the process to align the video, audio, and text modalities through a unified pre-training objective where we independently align the video and audio with the text modality.

*4.4.1 Contrastive Video-Text and Audio-Text Alignment:* Extending the contrastive video-text (VTC) loss proposed in ALPRO [18], we introduce audio-text (ATC) loss. The objective of the contrastive loss is to align the unimodal video and audio representations with their text counterparts. This forces unimodal encoders to produce embedding in the joint space of audio-text or video-text representation thereby facilitating the learning in multi-modal encoders.

For the set of video, audio, and text input features $< Y_{v,i}, Y_{a,i}, Z_{tm,i} >$ $m \in \{a, v\}$ in the batch, the similarity between two modality feature is defined as the dot product of the CLS tokens of the feature vectors. We are concerned about the alignment of the video and audio with the text modality. So, the similarity function of $i^{th}$ audio/video feature with the $j^{th}$ text feature is thus

$$s(Y_{m,i}, Z_{tm,j}) = m_{CLS,i} \cdot t_{CLS,j} \qquad (9)$$

where $m_{CLS,i} \in R^D, t_{CLS,j} \in R^D, m \in \{a, v\}$.

For each set of video, audio, and text features, the contrastive loss is expressed as the mean of the following two negative log-likelihood terms:

$$\mathcal{L}_{\text{m2t}} = -\log \frac{\exp\left(s\left(Y_{m,i}, Z_{tm,i}\right)/\tau\right)}{\sum_{j=1}^{B} \exp\left(s\left(Y_{m,i}, Z_{tm,i}\right)/\tau\right)}$$

$$\mathcal{L}_{\text{t2m}} = -\log \frac{\exp\left(s\left(Z_{tm,i}, Y_{m,i}\right)/\tau\right)}{\sum_{j=1}^{B} \exp\left(s\left(Z_{tm,i}, Y_{m,i}\right)/\tau\right)}$$

where $\tau$ is the temperature parameter which can be learnt, and $B$ is the batch size, and $m \in \{a, v\}$. The video-text and audio-text contrastive loss is the mean of the above two terms and can be expressed as:

$$\mathcal{L}_{\text{vtc}} = \frac{1}{2}\left(\mathcal{L}_{\text{v2t}} + \mathcal{L}_{\text{t2v}}\right) \qquad (10)$$

$$\mathcal{L}_{\text{atc}} = \frac{1}{2}\left(\mathcal{L}_{\text{a2t}} + \mathcal{L}_{\text{t2a}}\right) \qquad (11)$$

*4.4.2 Visual and auditory text matching:* The task involves visual-text matching and auditory-text matching, where the objective is to ascertain the similarity between pairs of videos or audio clips and corresponding textual descriptions. This process relies on multimodal encoders, denoted as $MME_{tv}$ and $MME_{ta}$, which produce joint representations of video-text and audio-text pairs, respectively. Specifically, these multimodal encoders generate modality pair-specific embeddings of the [CLS] token ($MM_{m,CLS}$, where $m \in \{a, v\}$) which act as comprehensive representations of the multimodal input. Subsequently, a fully connected layer coupled with softmax activation predicts the probability of a positive match for each pair, resulting in visual-text matching probability ($p^{\text{vtm}}$) and auditory-text matching probability ($p^{\text{atm}}$).

The loss functions for visual-text matching ($\mathcal{L}_{\text{vtm}}$) and auditory-text matching ($\mathcal{L}_{\text{atm}}$) are defined based on the cross-entropy between the predicted probabilities and the ground-truth labels -

$$\mathcal{L}_{\text{vtm}} = \mathbb{E}_{(V,T)\sim DS} \text{H}\left(\boldsymbol{y}^{\text{vtm}}, \boldsymbol{p}^{\text{vtm}}(V, T)\right) \qquad (12)$$

$$\mathcal{L}_{\text{atm}} = \mathbb{E}_{(A,T)\sim DS} \text{H}\left(\boldsymbol{y}^{\text{atm}}, \boldsymbol{p}^{\text{atm}}(A, T)\right) \qquad (13)$$

$\mathcal{L}_{\text{vtm}}$: Visual-Text Matching Loss, computed as the expected value over the dataset $DS$ of the cross-entropy between the ground-truth labels ($\boldsymbol{y}^{\text{vtm}}$) and the predicted probability distribution ($\boldsymbol{p}^{\text{vtm}}(V, T)$) for a video-text pair $(V, T)$.

$\mathcal{L}_{\text{atm}}$: Auditory-Text Matching Loss, also computed as the expected value over the dataset $DS$, representing the cross-entropy loss between the ground-truth labels ($\boldsymbol{y}^{\text{atm}}$) and the predicted probability distribution ($\boldsymbol{p}^{\text{atm}}(A, T)$) for an audio-text pair $(A, T)$.

In both equations, $\boldsymbol{y}^{\text{vtm}}$ and $\boldsymbol{y}^{\text{atm}}$ are 2-dimensional one-hot vectors representing the ground-truth labels, while $\boldsymbol{p}^{\text{vtm}}(V, T)$ and $\boldsymbol{p}^{\text{atm}}(A, T)$ denote the predicted probability distributions for visual-text and auditory-text pairs, respectively. The expectations are taken over the dataset $DS$, indicating the average loss over all samples in the dataset.

Furthermore, a strategy is employed to uncover challenging instances for both assignments, with the goal of refining training without increasing computational demands. These challenging instances, termed hard negatives, are pairs that exhibit comparable semantics but diverge in nuanced details. They are identified using metrics of contrastive similarity. Within each mini-batch, one adverse text/audio is selected based on the distribution of contrastive similarity, thereby increasing the likelihood of selecting texts/audios that closely resemble the video/audio under consideration. Moreover, for each text, one demanding video and audio pair is chosen, optimizing both visual-text and auditory-text alignment processes. This methodology enables the model to distinguish between positive instances and hard negatives, thereby enhancing its performance in both tasks.

## 4.5 Cumulative Pretraining Loss:

Contrastive loss (4.4.1) aims to align the unimodal encoders to produce embedding in a common representation space before passing the concatenated representation to the multimodal encoder. The matching loss (4.4.2) aims to align the multimodal encoder with the by learning to discern the positive video/text and audio/text pairs from the negative ones. The cumulative pre-training loss is thus expressed as the sum of contrastive loss and matching loss for the video-text and audio-text modality pairs.

$$\mathcal{L}_{\text{pre-training}} = \mathcal{L}_{\text{vtc}} + \mathcal{L}_{\text{atc}} + \mathcal{L}_{\text{vtm}} + \mathcal{L}_{\text{atm}} \qquad (14)$$

## 4.6 Downstream Fine-tuning:

With the aligned encoders, we fine-tune the model with the objective of complaint and aspect classification. Here the components explained in section 4.3 are utilized as in Figure 2, with the addition of two task-specific fully connected heads for simultaneously predicting the complaint and aspect classes. Each head comprises two linear layers with a softmax function finally producing the class probabilities of appropriate shape (2 for the complaint/non-complaint classification task and 7 for the aspect identification task). The loss function used in all tasks is categorical cross-entropy. The final loss function for the downstream multitask classification task ($Loss_{wmt}$) is a weighted sum of individual task-specific losses ($Loss_k$) for $M$ tasks, where the contribution of task $k$'s loss to the

overall loss is determined by the loss weight $\beta_k$ as shown in Equation (15).

$$Loss_{wmt} = \sum_{k=1}^{M} \beta_k Loss_k \qquad (15)$$

The parameters $\beta_i$ are learnt end-to-end, signifying task contribution from task $k$ to the multitask loss, enabling differential importance for parameter updates across tasks.

## 5 EXPERIMENTS AND RESULTS

### 5.1 Baselines

To analyze the contribution of different modality combinations and the effect of multi-tasking on the dataset, we established baselines involving individual modalities (uni-modal) and each pair of modalities (bi-modal).

In unimodal experiments, for text ($Z_{ta}$ or $Z_{tv}$), audio ($Z_a$), and visual ($Z_v$) modalities, we derived final representations using modality-specific encoders. These representations were then subjected to a CLS pooling operation, resulting in the extraction of overall semantic information contained within each modality. This extracted information is denoted as $m_{CLS}$, where $m$ belongs to the set $\{t, a, v\}$. Depending on single-task or multitask settings this was then passed through task-specific heads comprised of linear layers and finally through a softmax function to calculate the log probabilities for all tasks present in that setting. This setting is fine-tuned with the weighted aggregate of cross-entropy losses of all the present task-specific heads (15).

For bi-modal experiments, the representations obtained from the uni-modal encoders are concatenated and passed to the multimodal encoder to obtain the multimodal representation $C_m^s$ (6 and 7). Pooling operation is applied on the CLS token $MM_{m,CLS}$ of the multimodal representation and passed through task-specific heads as described previously. The losses are calculated in either single-task or multi-task manner and backpropagated.

The results for all the baseline experiments along with the proposed MAACA framework are mentioned in Table 3. Detailed analysis is conducted in the following sections.

### 5.2 Experimental Settings

All experiments were conducted on a machine equipped with an AMD EPYC 7552 48-Core Processor and 192 threads, coupled with 5 Nvidia A100 GPUs with VRAM memory of 40 GB per GPU card. For the experiments' preparation, the dataset was partitioned into testing, validation, and training sets at ratios of 15%, 15%, and 70%, respectively. To ensure robustness, the models were trained ten times with different random splits, and the average performance was reported. Hyperparameter configurations were tested rigorously, with the best results achieved using the Adam optimizer [14] and a learning rate set to $5e^{-5}$. The max text length of the tokenizer is set to 175 tokens which covers 95% of all the transcript lengths. A batch size of 2 is used, and all models are trained for 10 epochs with early stopping and patience of 2. All models were implemented in the PyTorch framework[3].

---

[3]https://pytorch.org/.

**Table 3: Results of various baselines involving different modality combinations on the VCD Dataset for the task of aspect category detection (ACD) and aspect complaint classification (ACC). The proposed MAACA framework leverages all three modalities. The bold results indicate the best-performing modality configuration for each task in Unimodal, Bimodal and, Trimodal settings.**

| Modality | MultiTask | ACC | | ACD | |
|---|---|---|---|---|---|
| | | Acc | F1 | Acc | F1 |
| **Unimodal** | | | | | |
| Video | No | 66.32 | 65.89 | 48.74 | 48.03 |
| | Yes | 67.21 | 66.70 | 49.93 | 49.25 |
| Audio | No | 59.67 | 58.72 | 31.18 | 30.27 |
| | Yes | 60.18 | 60.02 | 32.36 | 31.77 |
| Text | No | 84.49 | 83.82 | 83.55 | 82.91 |
| | Yes | **85.68** | **84.96** | **84.60** | **83.75** |
| **Bimodal** | | | | | |
| Video + Audio | No | 61.60 | 61.17 | 32.48 | 30.27 |
| | Yes | 62.34 | 61.92 | 34.59 | 33.86 |
| Text + Video | No | 86.27 | 85.81 | 86.29 | 85.72 |
| | Yes | 86.94 | 86.17 | **86.75** | **86.20** |
| Text + Audio | No | 87.05 | 86.24 | 85.63 | 84.97 |
| | Yes | **87.49** | **86.38** | 86.12 | 85.83 |
| **Trimodal** | | | | | |
| Text + Audio | No | 87.16 | 86.31 | 86.68 | 86.03 |
| + Video (MAACA) | Yes | **88.53** | **87.44** | **87.32** | **86.54** |

**Table 4: Ablation study to show the effect of removing pretraining step, moment retrieval, and multimodal gated fusion from the proposed model MAACA on both the tasks**

| Ablation Purpose | ACC | | ACD | |
|---|---|---|---|---|
| | Acc | F1 | Acc | F1 |
| **Proposed Framework** | **88.53** | **87.44** | **87.32** | **86.54** |
| - Pretraining | 84.32 | 83.89 | 83.61 | 82.45 |
| - Moment Retrieval | 87.50 | 86.92 | 85.21 | 84.32 |
| - Multimodal Gated Fusion | 86.09 | 85.31 | 84.59 | 83.19 |

### 5.3 Results and Discussion

Table 3 presents the main findings regarding the impact of different modalities on our analysis. Our uni-modal experiments reveal a significant discovery that forms the cornerstone of our proposed model: text (transcripts), contains the richest information for both the complaint (ACC) and aspect (ACD) tasks.

Building upon this, our bimodal experiments underscore the importance of text within the modality combinations. Models incorporating text alongside other modalities consistently outperform those relying solely on video and audio inputs. This highlights the pivotal role of textual data in our predictive framework.

Furthermore, our bimodal experiments shed light on another key insight: audio cues are crucial for complaint detection and visual information is conducive to aspect identification. This finding aligns with intuition: audio captures the reviewer's tone effectively, while videos provide a clearer depiction of the discussed aspects.

Consequently, these observations prompt us to integrate all three modalities—text, video, and audio—into our model to achieve optimal results across both complaint detection and aspect identification tasks. By leveraging the complementary strengths of each modality, we aim to enhance the robustness and effectiveness of our predictive framework. The optimal results were achieved on our proposed framework MAACA, in the multitask settings for both complaint and aspect tasks, amounting to 87.44% and 86.54% weighted F1 scores.

Comparing in the same task settings (single-task/multitask), the performance in Accuracy and weighted F1 scores follows a descending trend of Trimodal (MAACA), Bimodal, and, Unimodal configurations. This is further seen by the F1 score difference in just text and MAACA to be 2.48 % and 2.79 % respectively. As seen earlier Text and audio are better for complaint identification as in complaint data, there are more explicit acoustic markers in the audio as compared to the visual cues and MAACA leads Text + Audio in F1 scores by 1.04 %. As aspect identification is more visual cue-based, Text + Video performs on it better than Text + Audio, while MAACA still beats the first combination by an F1 score of 0.34 %. We further observe that across all modality combinations (Unimodal, Bimodal, and, Trimodal), multi-task models outperform single-task counterparts for most experiments. This can be intuitively explained by the fact that aspect and complaint identification are complementary tasks. Thus, the multitask setting aspect identification helps the framework in making an informed complaint/non-complaint classification decision, thereby boosting the quantitative metrics for both the classes.

*5.3.1 Ablation Study:* An ablation study is conducted to analyze the role of pre-training, moment retrieval, and multimodal gated fusion on the proposed framework MAACA. The findings are summarized in Table 4. As evident from the table, pre-training plays the biggest role in determining the performance of the model causing an increase of 3.55% in complaint F1 score and 4.09% in aspect F1 score. Pre-training aligns both the modality encoders and the multimodal encoders which results in better learning during the down-stream fine-tuning for the classification tasks. Following that, multi-modal gated fusion has a significant impact on the results. Instead of fusing the two multi-modal representations $C_v^s$ and $C_a^s$ to $J_{va}$ using the gating process (8), they are simply concatenated, and mean pooling is applied. The pooled representation is given as the input to the task heads and trained in a multi-task manner. Following the considerable decline in F1 score in both tasks, we see that a simple pooling of the multimodal representations is not as effective as the proposed gated attention mechanism. Lastly, moment retrieval's importance is tested in the ablation. From the results, we see the crucial role this pre-processing step plays in the identification of both the aspect and complaint causing an improvement of 0.52% and 2.22% in complaint and aspect F1 score respectively. Therefore, we highlight the need of pertaining, moment retrieval and multimodal gated fusion for the best model performance.

## 5.4 Limitations and Error Analysis

In this section, we examine certain challenges encountered by the proposed framework:

1. *Data Bias:* Given that the pre-training model ALPRO leveraged for the proposed work is trained on video-text corpus sourced from the web, it's susceptible to bias. This bias might manifest in the object detector, text, or video encoders, addressing this issue requires further analysis and training.

2. *Modality Restrictions:* The framework requires input from text, video, and audio data concurrently. If any modality is missing or incomplete, the model's predictive accuracy is significantly hindered.

3. *Incorrect Aspect Prediction:* In cases where the reviewer discusses a particular aspect with redundant information, the model fails to correctly identify the correct aspect class. This redundancy leads to inaccuracies in identifying the correct aspect class by the model. Additionally, the presence of redundant information can exacerbate the challenge of aspect prediction by introducing noise or irrelevant details that distract the model from discerning the essential aspect under consideration.

4. *Incongruent Complaint Tone:* We also observed that when the reviewer's audio tone doesn't align with the content conveyed through textual or visual modalities, leading to an incongruent complaint tone. In such cases, the reviewer's vocal expression or intonation may not accurately reflect the intent conveyed by the accompanying text or visuals. This inconsistency between the modalities confuses the model, making it difficult for it to accurately predict whether the instance should be classified as a complaint or not. For instance, the review "How have they still not managed to trim it at the bottom? Why has the selfie camera been demoted from 40 megapixels to just 12? Why are the colours of the phone so boring and samey?", the reviewer expresses complaint in an interrogative tone but the audio data point towards neutral tone, perhaps due to social norms, politeness, or other contextual factors.

## 6 CONCLUSION

In this research, we introduced a new challenge called Aspect-Level Complaint Detection from Discourse. The goal is to identify the category of aspects and determine whether they contain complaints or not, using the provided text, video, and audio data. To support the ACDD task, we created a distinctive multimodal complaint dataset named, VCD. This dataset underwent manual annotation, encompassing aspect categories and complaint/non-complaint labels based on textual, acoustic, and visual information extracted from video reviews. We conduct thorough analysis of the effect of various combinations of the modalities in predicting the aspect and complaint. We propose the MAACA framework to incorporate all three modalities as they are important for both the tasks. The results demonstrate that the inclusion of acoustic and visual features, in addition to text, enhances our ability to identify complaints at the aspect level. While the MAACA framework equipped with the proposed pre-training loss, fusion strategy, and moment retrieval step outperformed the baseline models, the error analysis highlighted that there is still room for improvement, which could be a focus for future research.

In the future, we aim to develop multimodal frameworks capable of pinpointing complaint rationales and incorporating them into a comprehensive summary beneficial for businesses. Additionally, we plan to extend our scope to include code-mixed videos and develop complaint-detection models tailored to such scenarios.

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
