# OpenReview forum: "Seeing Beyond Words: Multimodal Aspect-Level Complaint Detection in Ecommerce Videos"
_acmmm.org/ACMMM/2024/Conference — MM2024 Oral_

### Official Review · Reviewer_mTaE · 2024-05-21

**Rating:** 4
**Confidence:** 3

**Summary:**

This paper constructs a Video Complaint Dataset (VCD) and proposes a new task called Aspect Level Complaint Detection (ACDD). In addition, a multi-modal aspect-aware complaint analysis (MAACA) model is proposed, and the effectiveness of the model is verified through extensive experiments.

**Strengths:**

1. The paper is well-organized and clearly written.

2. The paper constructs a unique video complaint dataset (VCD), which will be of greater value for future research in this field.

3. The effectiveness of the proposed MAACA model has been demonstrated by extensive experiments.

**Limitations:**

1. The performance of the MAACA model seems to improve little relative to the baseline using text alone and the baseline using text and audio.
2. Figure 2 has some details that are not very clear and the connecting lines are not neat.
3. The presentation of the VCD dataset is a highlight of this paper, but the novelty of the MAACA model could be further enhanced.

**Suitability:**

3

---

### Official Review · Reviewer_LhFT · 2024-05-25

**Rating:** 3
**Confidence:** 2

**Summary:**

This paper proposes a multimodal aspect-aware complaint analysis framework for aspect-level complaint detection from discourse.

**Strengths:**

The task of complaint detection in ecommerce videos is novel and interesting.

**Limitations:**

1) The experimental comparison is missing. The effectiveness of the proposed framwork is difficult to evaluate.
2) Some case study should be shown for demonstrating the insight effectiveness of the roposed framwork.

**Suitability:**

3

---

### Official Review · Reviewer_LV94 · 2024-05-27

**Rating:** 4
**Confidence:** 3

**Summary:**

This paper proposes a unique multimodal product review dataset, which carefully annotates aspect categories and related complaint labels at the utterance level. And a multimodal aspect-aware complaint analysis model was introduced, providing valuable insights for the application of video-audio-text representation learning frameworks in downstream tasks.

**Strengths:**

1. The writing of this paper is excellent and easy to understand.

2. The construction of datasets is a very heavy workload and has a significant promoting effect on this field.

3. The code and data are open source, providing replicability.

**Limitations:**

1. Lack of innovation in methods. From multimodal fusion to the design of pre-training tasks, there are existing works that lack specificity for this task.

2. The experiment lacks replication of baseline methods. For example, many methods in the field of emotion recognition in conversations are applicable.

3. Lack of case studies. It would be even better if there were visualization experiments to illustrate the model.

**Suitability:**

3

---

### Meta-Review · Area_Chair_BoUh · 2024-06-26

**Recommendation:** Accept (Oral)
**Confidence:** 4

**Metareview:**

The reviewers have come up to the following strengths and limitations

STRENGTH
- Clarity and Accessibility
- Dataset Construction
- Open Source Code and Data
- Novelty and Interest
- Organization and Coherence
- Unique Contribution with VCD
- Demonstrated Effectiveness of MAACA Model

LIMITATION
- Lack of Methodological Innovation
- Fail to Replication of Baseline Methods
- Absence of Case Studies and Visual Experiments
- Limited Performance Improvement of MAACA Model
- Figure Clarity Issues
- Mixed Emphasis on Novelty

The rebuttal did well, reviewers agreed to raise their rating.